# Estrus Detection and Optimal Insemination Timing in Holstein Cattle Using a Neck-Mounted Accelerometer Sensor System

**DOI:** 10.3390/s25175245

**Published:** 2025-08-23

**Authors:** Jacobo Álvarez, Antía Acción, Elio López, Carlota Antelo, Renato Barrionuevo, Juan José Becerra, Ana Isabel Peña, Pedro García Herradón, Luis Ángel Quintela, Uxía Yáñez

**Affiliations:** 1Unit of Reproduction and Obstetrics, Department of Animal Pathology, Faculty of Veterinary Medicine, Universidade de Santiago de Compostela, Avda. Carballo Calero s/n, 27002 Lugo, Spain; jacobo.alvarez.torres@rai.usc.es (J.Á.); antia.accion.carro@rai.usc.es (A.A.); renato.barrionuevo@rai.usc.es (R.B.); ana.pena@usc.es (A.I.P.); luisangel.quintela@usc.es (L.Á.Q.); uxia.yanez.ramil@usc.es (U.Y.); 2Innogando, A Graña s/n, Abadín, 27730 Lugo, Spain; elio@innogando.com (E.L.); carlota@innogando.com (C.A.); 3Instituto de Biodiversidade Agraria e Desenvolvemento Rural (IBADER), Universidade de Santiago de Compostela, Lugo Campus, 27002 Lugo, Spain

**Keywords:** dairy cattle, heat detection, precision livestock farming, reproductive performance, ultrasonography

## Abstract

**Highlights:**

**What are the main findings?**
The neck-mounted accelerometer-based sensor RUMI demonstrated high specificity, sensitivity, and accuracy for estrus detection in high-producing Holstein cows.The optimal timing for artificial insemination was identified as 11–15 h after estrous onset.

**What is the implication of the main findings?**
Improved heat detection accuracy can enhance reproductive performance and reduce economic losses associated with missed estrus events and failed inseminations.The integration of sensor-based estrus detection reduces reliance on hormones, addressing societal concerns regarding animal welfare and food safety.

**Abstract:**

This study aimed to evaluate the accuracy of the accelerometer-equipped collar RUMI to detect estrus in dairy cows, establish a recommendation for the optimal timing for artificial insemination (AI) when using this device, and characterize the blood flow of the dominant follicle (F) and the corpus luteum (CL) as ovulation approaches. Forty-seven cycling cows were monitored following synchronization with a modified G6G protocol, allowing for spontaneous ovulation. Ultrasound examinations were conducted every 12 h, starting 48 h after the second PGF2α dose, to monitor uterine and ovarian changes. Blood samples were also collected to determine serum progesterone (P4) levels. Each cow was fitted with a RUMI collar, which continuously monitored behavioral changes to identify the onset, offset, and peak of activity of estrus. One-way ANOVA assessed the relationship between physiological parameters and time before ovulation. Results showed that the RUMI collar demonstrated high specificity (100%), sensitivity (90.90%), and accuracy (93.62%) for estrus detection. The optimal AI window was identified as between 11.4 and 15.5 h after heat onset. Increased blood flow to the F and reduced luteal activity were observed in the 48 h prior to ovulation. Further research is needed to assess the influence of this AI window on conception rates, and if it should be modified considering external factors.

## 1. Introduction

Inefficient heat detection remains a persistent challenge in dairy farms, driven by a range of biological and management-related factors [1]. The selection of high-producing cows (HC) has been associated with decreased expression of estrus behavior and increased incidence of silent estrus [2,3,4]. At the same time, HC tend to exhibit shorter estrus periods, probably due to lower circulating estradiol levels [2,5]. Reduced labor availability in dairies is a problem that arose in the last century [6]. This has limited the ability to detect estrus by visual observation due to the requirement of several periods of daily intensive observation and increasing herd sizes [7,8]. Traditional estrus detection methods, such as visual observation, tail chalking, and scratch-off patches, have proven insufficient under modern farming conditions [6]. Currently, HC often display fewer primary signs of estrus, including standing to be mounted, while exhibiting more secondary behaviors like mounting, butting, sniffing, licking, and chin-resting [8]. However, some authors have associated these behavioral changes with housing conditions [8].

Deficiency in heat detection has driven the adoption of estrus-synchronization protocols and the use of fixed-time artificial insemination (FTAI), by combining treatments based on gonadotropin-releasing hormone (GnRH) and prostaglandin F_2α_ (PGF_2α_) [9,10]. These protocols use hormones to control follicular development and corpus luteum (CL) regression [11,12]. Synchronization protocols in combination with FTAI make heat detection unnecessary [9]. However, increasing consumer concerns about food safety and hormone use in livestock have prompted the search for non-hormonal alternatives [13]. Therefore, during the last two decades, new heat-detection methods have been developed, such as video cameras, activity monitoring systems, measurement of body temperature, and recording vocalization [8].

The most used methods nowadays are activity-monitoring systems, including pedometers and neck-mounted accelerometer-equipped collars, among others [8,14]. These systems are based on the monitoring of physical activity, which increases during heat [14]. This is due to the fact that during estrus, the high concentration of estradiol (E2) increases physical activity in cows [6,14,15]. Several research papers analyzing ovulation time and conception rate after the increase in physical activity have been published [6,15,16,17,18]. At the same time, during heat, P4 blood concentration decreases up to basal levels and can be measured to observe CL regression [11,19,20,21]. Other studies found that during estrus, ovarian structures vary significantly in comparison to animals in proestrus [20,21,22,23]. Likewise, the preovulatory follicle increases its blood flow around the peripheral area as ovulation approaches [24]. Therefore, changes in ovaries could be detected by ultrasound [19,21,23].

Pedometers were the first devices used, serving both to monitor activity and identify individual cows [15]. However, the accuracy of this system is diminished in farms with small dimensions and little space available per cow [15]. Additionally, it is necessary to distinguish between the peak of activity caused by heat and the increasing number of steps due to other circumstances, such as unusual managing events [15]. In contrast, neck-mounted accelerometer-equipped collars provide more comprehensive data, including individual activity, rumination, feeding, and resting patterns, and send this information to the designated device or program [16], which analyses these raw data using an algorithm. Based on the differences between the current information and previous data, the algorithm is able to determine substantial behavioral changes. As estrus is characterized by an increase in physical activity and a decrease in rumination and feeding times. These devices can identify cows in heat according to the information they have collected.

As it has previously been described, a poor heat-detection rate is one of the most significant economic loses in dairies, reaching up to $360 per missed estrus event [6]. Likewise, heat detection is the most important factor impacting the success of artificial insemination (AI) [4]. If AI occurs too late after the onset of estrus, spermatozoa will not have enough time for capacitation, causing the absence of fecundation or the fertilization of old oocytes, resulting in low quality embryos [1]. However, if AI occurs too early, many spermatozoa will die before the oocyte arrives at the oviduct [1]. Therefore, accurate prediction of ovulation is a critical factor of a successful estrus-detection program, and it should ideally be validated using gold-standard methods such as ultrasonography [17]. In this context, the following objectives were established: (1) to evaluate the accuracy of an accelerometer-equipped collar (RUMI, Innogando, Spain) to detect estrus in dairy cows; (2) to establish a recommendation for the optimal timing for artificial insemination when using this particular device; and (3) to characterize the blood flow of the dominant follicle and the corpus luteum as ovulation approaches.

## 2. Materials and Methods

### 2.1. Animals

A total of 47 Holstein cows were enrolled in the study. They were housed at “Granxa Experimental Campus Terra” (Castro de Rei, Spain), a free-stall facility with rice-husk-based silica/carbon-bedded cubicles. Cows were milked 3.2 times/day with an automatic milking machine (daily milk production 40.44 kg/cow), were fed a total mixed ration once a day and had *ad libitum* access to water. Reproductive examinations, including pregnancy diagnosis and postpartum evaluations, were carried out twice a month by an experienced veterinarian. Reproductive information was recorded on the farm software Gando version 2024 (Gando Nuevas Tecnologías, Ferrol, Spain). All cows enrolled were equipped with RUMI monitoring devices (Innogando, Lugo, Spain) that recorded real time information about their activity and heat detection.

### 2.2. Automated Estrus-Detection Systems

RUMI (Innogando, Lugo, Spain) is an intelligent device that is placed on a collar on the left side of the animal’s neck designed for monitoring the activity and well-being of livestock. The RUMI device is equipped with a triaxial accelerometer and a GPS module for continuous behavioral and location monitoring of cattle. The accelerometer samples at a rate of 20 Hz (typical range for livestock behavior-monitoring devices: 10–25 Hz), while GPS positions are updated at configurable intervals to optimize data resolution and battery efficiency. Data are transmitted via a proprietary long-range radio link to a local base station, which forwards the information to a cloud-based server for processing.

The device is designed for outdoor agricultural environments, with an IP67-rated enclosure, an operating temperature range from −20 °C to +50 °C, a weight of approximately 250 g, and an exceptionally long battery life of up to 7 years (depending on configuration and usage).

This monitoring device allows for the real-time detection of various behavioral parameters from the animals: resting, ruminating, grazing, walking, and activity. RUMI is equipped with motion and accelerometer sensors, combined with behavioral analysis algorithms designed to identify characteristic patterns in each monitored parameter for individual cows. The RUMI estrus detection algorithm integrates multiple behavioral parameters—including activity levels, movement patterns, and spatial data—using an artificial intelligence model trained on extensive historical datasets. Each parameter contributes to an overall estrus index, with thresholds established through experimental calibration. Specific parameter weightings and threshold values constitute proprietary elements of the system and cannot be disclosed in full detail, as RUMI is a commercial product.

### 2.3. Synchronization of Estrus

In the current study, only cows with more than 60 days in milk that had resumed cycling ovarian activity after the last calving were included. Following Yáñez et al.’s [20] study design, a modified ovulation synchronization protocol, G6G, with 2 (G6G1PG) or 3 (G6G2PG) PGF2α administrations (150 μg of PGF2α analogue Dinoprost, Enzaprost^®^ T, Ceva Salud Animal S.A., Barcelona, Spain) was used (Figure 1). The final GnRH was not administered to allow for spontaneous ovulation. Forty-eight hours after the second PGF2α, ultrasound examinations were carried out twice a day until ovulation, which was considered time 0. Heat information (onset, conclusion of heat, and maximum heat) were retrospectively obtained from the activity monitoring systems.

### 2.4. Ultrasound and Sampling

A ProVetScan SR-2C (New Veterinary Technologies, León, Spain), equipped with a multifrequency (6.5–8 MHz), linear-array transducer (Frequency 8.0 MHz, Gain 56%, PRF 2.0 K) was used to perform B-mode and Power Doppler examinations. All examinations were recorded and analyzed by the same person. Uterus measurements (endometrium thickness (END), myometrium and perimetrium thickness (MIO), and uterine lumen (UL)) and diameter of the dominant follicle (F) and corpus luteum (CL) were collected. If two Fs or two CLs were observed in the same cow, measurements were recorded for both structures. To distinguish between them, letters A or B were assigned (e.g., FA or FB and CLA or CLB). MIO and END were measured just before the curvature of the uterine horns. Ratio END/MIO was calculated as an index to contemplate END and MIO measurements as they, respectively, increased and decreased during the period of study. The UL was measured at the maximum content localization. Ovulation was confirmed by the disappearance of the F in two contiguous examinations, and ovulation time was defined as the time of the last examination minus 6 h (mid-time between two subsequent examinations). Additionally, activity information of the F and the CL were recorded with Power Doppler examinations. A subjective evaluation was performed, classifying F blood flow as absent (0) or present (1, 2, and 3 as it gradually increased) according to previous research [20]. CL blood flow was classified as absent (0) or present (1, 2, and 3 as it gradually increased).

Blood samples were collected from the coccygeal vein using vacuum tubes without anticoagulants. Samples were kept in refrigeration until the blood clot was formed. Serum was separated from the clot by centrifugation for 10 min at 1500× *g* and was stored at −18 °C into 1 mL reaction tubes until analysis.

Serum progesterone (P4) concentrations were measured with a commercial progesterone ELISA kit (Progesterone ELISA EIA-1561, DRG Instruments GmbH, Marburg, Germany) following the manufacturer’s instructions. The detection limit for P4 ranged from 0 to 40 ng/mL, with an analytical sensitivity of 0.045 ng/mL. Optical densities were measured in a microplate reader (Multiskan EX, Thermo Fisher Scientific Inc., Waltham, MA, USA).

### 2.5. Statistical Analyses

A one-way ANOVA test was performed including ratio END/MIO, END, MIO, UL, P4, maximum and minimum diameter of the F and CL as dependent variables and the time before ovulation (−48, −36, −24, −12 and −6 h) as a factor. The confidence interval was obtained for heat offset and heat onset. Cross tables were performed, including protocol response and subjective blood flow evaluation as columns and heat parameters, and P4 and interval PGF2α–ovulation as rows. The Bonferroni test was used to perform post-hoc comparisons, the homogeneity of variances was checked using Levene’s test, and normality was tested using kurtosis and asymmetry (values ranging from −0.5 to 0.5).

All analyses were conducted in IBM SPSS Statistics version 28.0 for Windows (IBM Corp., Armonk, NY, USA). Differences were considered significant at *p* ≤ 0.05.

## 3. Results

Following the final administration of PGF2α, 70.2% (G6G1PG: 73.3%; G6G2PG: 64.7%) of the enrolled animals responded to treatment (Table 1). No statistically significant differences were observed between the two synchronization protocols for any of the variables analyzed (*p* > 0.05; Table 1).

As displayed in Table 2, 14 cows were diagnosed as not being in heat by ultrasonography and RUMI at the same time. Out of the 33 cows diagnosed as being in heat by ultrasound examination, 30 were detected by RUMI. Therefore, RUMI had a sensitivity of 90.90% for heat detection. All cows diagnosed as not in heat by ultrasonography were consistently classified as not in heat by RUMI as well, resulting in 100% specificity. In general, RUMI monitoring devices had an accuracy of 93.62%.

Heat characteristics were analysed (Table 3), focusing on heat duration, heat intensity, various intervals (at 48, 36, 24, 12, and 6 h before ovulation) related to ovulation and P4 levels. The average duration of heat was 11.90 ± 4.34 h. The maximum heat intensity was recorded at 6.27 ± 4.05 h from the heat onset. The interval between the second administration of PGF2α and ovulation was 90.80 ± 19.85 h (Figure 2). Finally, as depicted in Figure 3, 68.97% of the cows had less than 1 ng/mL P4 41 h before ovulation.

Heat onset was observed during the night, morning, and afternoon in 36.7%, 30.0%, and 33.3% of cows, respectively. As can be seen in Figure 4, the intervals between heat onset and offset and ovulation were 27.47 ± 5.42 and 15.30 ± 6.99 h, respectively. The 95% confidence interval for heat onset–ovulation and heat offset–ovulation intervals was 25.4–29.5 h and 12.70–17.90 h, respectively.

As ovulation approached, the diameter of the FA and the FB increased in maximum and minimum diameter from 48 to 6 h before ovulation. However, size increase and maximum and minimum diameter were only significantly bigger in the FA, as displayed in Table 4. Neither the maximum nor the minimum CL diameters significantly varied from 48 h before ovulation. At the same time, uterus measurements did not significantly vary from 48 h before ovulation. On the other hand, P4 measurements had a significant diminishment from 48 (0.81 ± 1.07 ng/mL) to 6 (0.27 ± 0.32 ng/mL) hours before ovulation.

Power Doppler evaluations showed that, as ovulation approached, the F had an increase in blood flow, which was significant only in the FA. At the same time, Power Doppler showed that, as the FA had more blood flow, the CL had less vascularization until 6 h before ovulation, when no CLs had blood flow detected by Power Doppler (Table 5).

## 4. Discussion

Our results showed that RUMI monitoring devices had an overall accuracy of 93.6%, with a sensitivity and specificity of 90.9% and 100%, respectively, to detect Holstein cows in heat. These figures suggest that RUMI’s ability to detect heat is higher than that of other devices used in previous studies, in which only 37%, 70%, and 80% of cows in estrus were correctly identified [13,15]. One of the possible explanations for these differences is that RUMI monitoring devices use an algorithm based on several behavioral parameters from the last week of the cow itself, while devices used in other studies only take into account the accelerometer sensor information. However, it should be noted that similar accuracy (99.0%) has been previously reported for accelerometers, either alone or combined with augmented reality and deep learning models [9,25,26]. Despite the similarities regarding sensitivity, the specificity provided by RUMI devices was 100%, while other heat detection systems were reported to show false estrus events [25,26,27,28]. A recently developed device based on sensor data achieved 96% accuracy in heat detection; however, it was tested on just five cows [29]. Another data analysis tool for processing information from automated activity monitoring systems showed an error rate of 14.54% in heat detection [30]. Likewise, a different device reached an accuracy of 89.50% in detecting estrus [31]. This difference may be explained by the fact that the latter device recorded data at a frequency of 10 Hz, whereas RUMI records at 20 Hz [31]. As for heat characteristics, the cattle enrolled in this study showed an even distribution of heat onset throughout the day, similarly to previous research [32]. Additionally, heat duration observed was in accordance with other studies reporting an average heat duration of 14 h (range 8–20 h, [26]). It should be noted that shorter heat duration (6.2 h, range 0.4–26.5) has been described in the literature for high milk producing cows, which may be justified by the higher metabolic clearance rate of P4 and estradiol [2]. In our study, although ultrasound examinations were performed every 12 h, it has been described in the literature that this is not likely to alter estrus behavior [33].

Regarding the heat onset–ovulation interval (25.4–29.5 h, 95% CI), similar results have been observed [22,24]. In fact, the latter review reported information collected from nine different papers, in which ovulation occurred between 24 and 33 h after heat onset. Concerning the heat offset–ovulation interval (12.7–17.9 h, 95% CI), the same review described an average range of 15–21 h [34]. In our study, maximum heat activity was detected 6.27 h (1–20) after heat onset. This finding partially agrees with another study, where the highest increase in heat activity (34.2%) was 8 h before to 5 h after heat onset [8]. Differences could be due to the high variation among cows to display maximum activity during heat [8].

One noticeable difference between our study and other studies published on this topic is the use of a modified FTAI protocol to synchronize cows. This protocol was routinely used in the dairy farm where the study was conducted, and it was chosen to interfere as little as possible with normal management. A response of 70.2% was obtained, similar to what was found in previous research (67.1%) for the same protocol [35]. It should be also noted that no differences were observed between the use of 2 or 3 PGF_2α_ administrations, in accordance with Heidari et al. [36], who reported a response of 72.0% and 74.0%, respectively. Additionally, no differences in estrus characteristics were observed regarding the protocol, as also described by Heidari et al. [36]. The use of 1 or 2 PGF_2α_ administrations prior to the last GnRH in FTAI protocols has been a topic of discussion for years, with still pending discrepancies. Contrary to what we described, some researchers reported significant differences regarding CL regression between the two treatment groups, observing a better response for the two doses and an increased pregnancy/AI ratio and CL regression [21]. Carvalho et al. [21] concluded that this observation could be the consequence of an incomplete regression of young CL (seven days after ovulation) after a single administration.

Considering uterine measurements, no significant variations were observed after 48 h before ovulation. This finding contrasts with the results reported in previous research, where it was concluded that the appearance of the uterus varies during the estrus cycle and uterine shape and intrauterine fluid increases around ovulation time [37]. This could be explained because this research group analyzed data collected at larger intervals of time (days) than we did (hours). As for ovarian structures, neither the maximum nor the minimum diameter of the CL significantly varied during the 48 h before ovulation. Additionally, no CL was classified as active using Power Doppler ultrasound or progesterone concentrations. This could be explained by the fact that the first examination was conducted two days after PGF_2α_, giving enough time to allow for CL regression, with the subsequent decrease in P4 concentration and start of the follicular phase. In addition, the diameter of the F was within the range reported in previous studies in Holstein cattle [38]. The higher diameter compared to other breeds, such as beef cows [26], could be due to the higher liver blood flow and subsequent metabolism of steroid hormones in high producing breeds [39]. Additionally, the F tended to increase its vascularization, measured by Power Doppler ultrasound, as ovulation approached, in accordance with research that also reported higher blood flow around the peripheral area of the preovulatory follicle [35,40]. This increase in blood flow could be a consequence of the LH surge previous to ovulation [23], and it could be used as a reference to estimate the expected time of ovulation in a more precise way. However, although assessing the blood flow to the F may be useful within the scope of research, its use in commercial farms is impractical due to the need for repeated ultrasound examinations.

When the last PGF_2α_ of the protocol was administered, ovulation occurred 90.8 h (52–136) later. Similar information has been previously reported, with ovulation occurring 82.2 h after the induction of luteolysis [15]. A recent study that evaluated two different analogues and doses of PGF_2α_, found that Holstein heifers ovulated 69.3 ± 1.3, 71.3 ± 1.3, and 72.9 ± 1.7 h after the administration of the PGF_2α_ analogue [41]. The small time variations could be explained by the differences in the protocols used to induce ovulation. For instance, the protocol used in this study, G6G, includes a pre-synchronization phase that was not used in other studies [15,41]. This pre-synchronization phase aims to prepare the animal for the subsequent hormone administrations, the actual Ovsynch protocol, and it has been shown to increase the ovulatory response [42]. Further research should be conducted to study the influence of different breeds on ovulatory response and ovulation intervals.

Following recommendations to perform AI 14 h before ovulation to achieve better conception rates [43], cows monitored with RUMI should ideally be inseminated ~13.5 h (95% CI: 11.4–15.5) or ~1.3 h (95% CI: −1.3–3.9) after heat onset and heat offset, respectively. However, the study design may have resulted in an estimated error of approximately 6 h in ovulation timing for some cows, due to ultrasound examinations being performed at 12 h intervals, which constitutes a limitation of the current study. This insemination window is narrower than the optimal time described by De Rensis et al. [34], who concluded that cattle should be inseminated 4 to 18 h after heat onset and 6 to 8 h after heat offset. Further research should be conducted to check the effect of this insemination window regarding conception rates, as it has been described that the timing of AI, using conventional semen, did not significantly influence conception rates if it was performed between 0 and 32 h from the onset of estrus and −12–20 h from the end of estrus [44]. Nevertheless, the same researchers found different results for sex-sorted semen, observing better conception rates when AI was performed between −4 and 12 h from the end of estrus. Nevertheless, Furukawa et al. [44] used a general period of ovulation of 12 h after the end of estrus in every cow used in their study. In contrast, another study found that there were no significant differences in the optimal time of insemination to conventional semen and the conception rate [45]. These findings highlight the importance of external factors, such as the higher sensitivity of sperm exposed to sex-sorting regarding the outcome of AI, and how they should be included in the equation for each system to provide an optimal AI window for this type of semen. Moreover, differences observed among studies suggest that each monitoring device should be carefully validated and information about heat detection and optimal AI time must not be extrapolated to different devices.

## 5. Conclusions

The neck-mounted accelerometer-equipped collar RUMI has high specificity (100%) and sensitivity (90.9%) for estrus detection in Holstein cattle, with an accuracy of 93.6%. Additionally, AI should be performed 13.5 h after heat onset, with an optimal AI window of 11.4 to 15.5 h. However, further research is needed to check the influence of this AI window on conception rates, and also if it should be modified considering external factors, such as the use of sex-sorted semen, breed, and management.

## Figures and Tables

**Figure 1 sensors-25-05245-f001:**
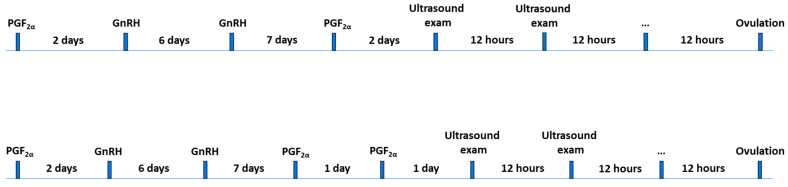
Schedule of the study design, including hormone administrations following the modified synchronization protocol, G6G, with 2 (above) or 3 (below) prostaglandin F2α (PGF2α) administrations, and the ultrasound examinations until ovulation. GnRH: gonadotropin-releasing hormone.

**Figure 2 sensors-25-05245-f002:**
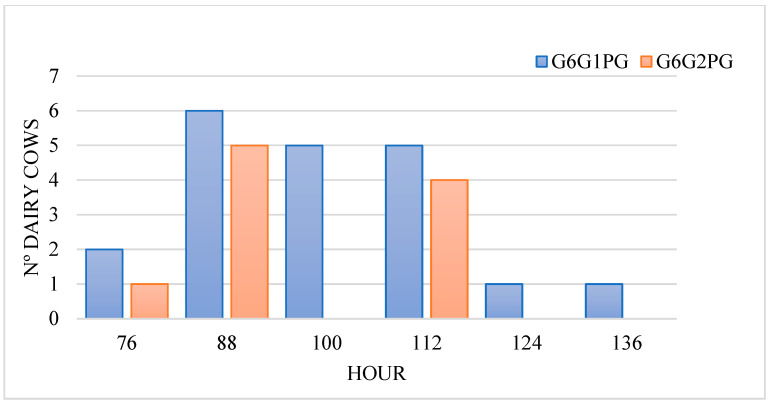
Distribution of Holstein cows according to the interval between the second administration of prostaglandin F2α (PGF2α) and ovulation time for the two modified synchronization protocols used in the study: G6G with 2 (G6G1PG) or 3 (G6G2PG) PGF2α administrations.

**Figure 3 sensors-25-05245-f003:**
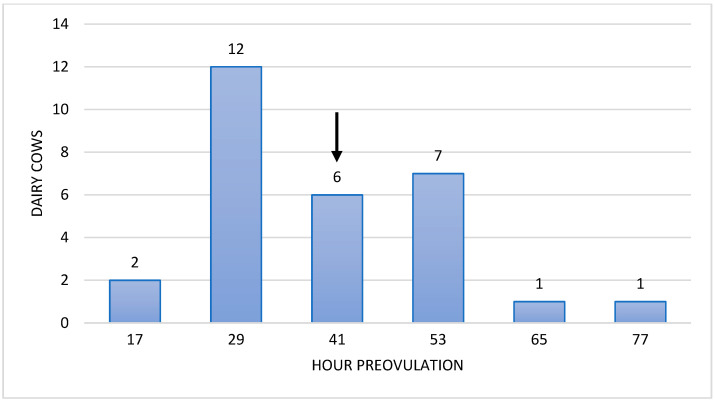
Distribution of Holsteins cows according to the time before ovulation when the serum progesterone levels (ng/mL) were lower than 1 ng/mL. The arrow indicates the median.

**Figure 4 sensors-25-05245-f004:**
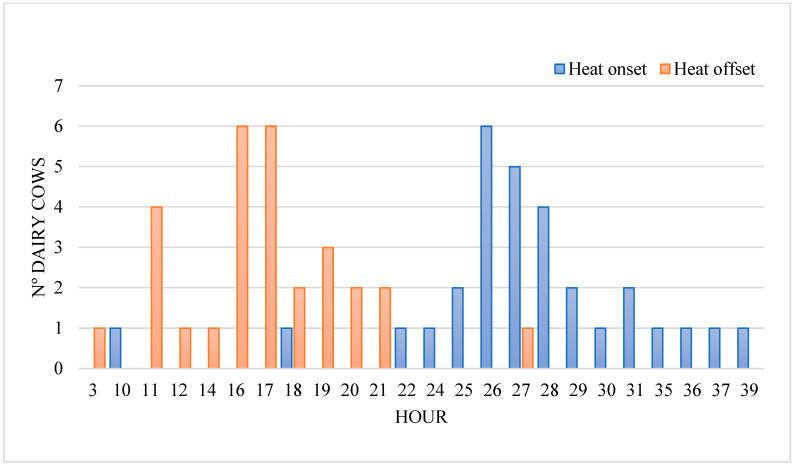
Distribution of 33 Holstein cows according to the heat onset–ovulation and heat offset–ovulation intervals.

**Table 1 sensors-25-05245-t001:** Descriptive data (mean ± SD) for heat characteristics according to the synchronization protocol used. PGF2α: Prostaglandin F2α; G6G1PG: Modified G6G with 2 PGF2α administrations; G6G2PG: Modified G6G with 3 PGF2α administrations.

Protocol	Response	Heat Duration (h)	Maximum Heat (h)	Interval Heat Onset–Ovulation (h)	Interval Heat Offset–Ovulation (h)	Interval Last PGF_2α_–Ovulation (h)	Time Before Ovulation When Progesterone < 1 ng/mL (h)	Interval 2nd PGF_2α_–Ovulation (h)
G6G1PG	73.33% (22/30)	12.40 ± 4.51	6.80 ± 4.54	28.45 ± 6.05	15.80 ± 7.76	100 ± 15.57	41.00 ± 15.57	100.00 ± 15.57
G6G2PG	64.70% (11/17)	10.90 ± 4.01	5.20 ± 2.74	25.50 ± 3.34	14.30 ± 5.36	72.40 ± 13.91	35.67 ± 10.58	96.4 ± 13.91

**Table 2 sensors-25-05245-t002:** Accuracy of accelerometer-equipped collar RUMI in diagnosing heat in Holstein cows compared to ultrasonography examinations.

RUMI Alert	Ultrasonography	
	Heat	No Heat
Heat	90.9% (30/33)	0% (0/14)
No heat	9.1% (3/33)	100% (14/14)

**Table 3 sensors-25-05245-t003:** Heat characteristics for 33 Holstein cows provided by the accelerometer-equipped collar RUMI. h: hours.

	Mean ± SD	Minimum	Maximum	Median
Heat duration (h)	11.90 ± 4.34	6	21	11.50
Maximum heat (h)	6.27 ± 4.05	1	20	6
Interval heat onset–ovulation (h)	27.47 ± 5.42	10	39	27
Interval heat offset–ovulation (h)	15.30 ± 6.99	14	27	17
Interval 2nd PGF2α–ovulation (h)	90.80 ± 19.85	52	136	88
Time before ovulation when progesterone < 1 ng/mL (h)	39.34 ± 14.24	17	77	41

**Table 4 sensors-25-05245-t004:** Evolution during the 48 h before ovulation of serum progesterone (P4) levels and ultrasound measurements: uterine measurements (endometrium thickness (END), myometrium and perimetrium thickness (MIO), ratio END/MIO and uterine lumen (UL)); maximum (MFA) and minimum (mFA) diameter of the first dominant follicle; maximum (MFB) and minimum (mFB) diameter of the second dominant follicle; maximum (MCL) and minimum (mCL) diameter of the CL.

Hours Before Ovulation	−48 (n = 11)	−36 (n = 17)	−24 (n = 26)	−12 (n = 30)	−6 (n = 30)
P4 (ng/mL)	0.81 ± 1.07 a	0.42 ± 0.44 ab	0.32 ± 0.20 b	0.39 ± 0.37 ab	0.27 ± 0.32 b
MFA (mm)	16.80 ± 2.57 a	19.41 ± 4.16 ab	20.33 ± 4.52 ab	21.14 ± 3.90 b	20.89 ± 4.57 b
mFA (mm)	13.89 ± 2.42 a	15.74 ± 3.40 ab	17.11 ± 4.02 ab	16.56 ± 2.79 ab	17.21 ± 3.02 b
MFB (mm)	14.62 ± 7.10 a	17.60 ± 4.03 a	17.86 ± 4.10 a	18.28 ± 4.12 a	17.33 ± 5.14 a
mFB (mm)	12.67 ± 5.81 a	14.21 ± 4.11 a	14.32 ± 3.21 a	14.68 ± 4.64 a	13.86 ± 3.16 a
MCL (mm)	18.41 ± 3.20 a	17.84 ± 4.69 a	17.03 ± 2.86 a	17.60 ± 3.61 a	17.53 ± 1.80 a
mCL (mm)	14.14 ± 3.20 a	13.97 ± 3.35 a	14.00 ± 1.69 a	13.93 ± 3.15 a	13.70 ± 2.83 a
END (mm)	10.49 ± 2.17 a	9.70 ± 2.41 a	10.18 ± 3.33 a	9.81 ± 2.86 a	9.93 ± 2.64 a
MIO (mm)	4.39 ± 1.10 a	4.73 ± 1.81 a	3.83 ± 0.89 a	4.42 ± 1.39 a	4.76 ± 2.30 a
END/MIO	2.52 ± 0.75 a	2.39 ± 1.46 a	2.79 ± 1.16 a	2.40 ± 0.89 a	2.36 ± 0.90 a
UL (mm)	6.21 ± 4.73 a	6.26 ± 2.95 a	5.05 ± 4.19 a	5.04 ± 3.85 a	3.44 ± 3.80 a

Different letters indicate significant differences (*p* ≤ 0.05) within the same row.

**Table 5 sensors-25-05245-t005:** Subjective blood flow evaluation of the first dominant follicle (FA), the second dominant follicle (FB) and the corpus luteum (CL) during the 48 h before ovulation.

Hours Before Ovulation	−48	−36	−24	−12	−6
FA *	N = 10	N = 16	N = 22	N = 27	N = 29
0	100% (10/10)	81.3% (13/16)	86.4% (9/22)	70.4% (19/27)	51.7% (15/29)
1	0% (0/10)	18.75% (3/16)	9.1% (2/22)	3.7% (1/27)	20.6% (6/29)
2	0% (0/10)	0% (0/16)	0% (0/22)	25.9% (7/27)	13.8% (4/29)
3	0% (0/10)	0% (0/16)	4.5% (1/22)	0% (0/27)	13.8% (4/29)
FB *	N = 5	N = 11	N = 14	N = 14	N = 14
0	100% (5/5)	90.9% (10/11)	92.9% (13/14)	78.6% (11/14)	85.7% (12/14)
1	0% (0/5)	9.1% (1/11)	0% (0/14)	7.1% (1/14)	0% (0/14)
2	0% (0/5)	0% (0/11)	0% (0/14)	14.3% (2/14)	7.15% (1/14)
3	0% (0/5)	0% (0/11)	7.1% (1/14)	0% (0/14)	7.15% (1/14)
CL **	N = 8	N = 12	N = 10	N = 10	N = 7
0	12.5% (1/8)	66.67% (8/12)	70% (7/10)	80% (8/10)	100% (7/7)
1	50% (4/8)	16.67% (2/12)	20% (2/10)	20% (2/10)	0% (0/10)
2	37.5% (3/8)	16.67% (2/12)	10% (1/10)	0% (0/10)	0% (0/10)

* A subjective evaluation was performed, classifying F blood flow as absent (0) or present (1, 2, and 3 as it gradually increased) according to previous research [20]. ** CL blood flow was classified as absent (0) or present (1, 2, and 3 as it gradually increased).

## Data Availability

Data can be provided by the correspondence author under reasonable request.

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
