# Peer review of "Estrus Detection and Optimal Insemination Timing in Holstein Cattle Using a Neck-Mounted Accelerometer Sensor System"

_sensors, 2025, doi:10.3390/s25175245_

Round 1

Reviewer 1 Report

Comments and Suggestions for Authors

The work is very contemporary, focusing on precision dairy cattle breeding and the contributions RUMI can offer farmers in heat detection and optimal insemination timing. The most significant objection I find to the work is the second proposed objective: "To determine the optimal timing of artificial insemination using the device."

The methodology followed does not allow for determining the optimal timing of insemination, since artificial insemination is not performed. However, it does allow for establishing a recommendation for the optimal insemination interval, based on one's own observations and the results of other authors. I believe the objective should not be "TO DETERMINE" but rather "TO ESTABLISH a recommendation for the optimal insemination interval."

The use of the device facilitates identifying this timing. Fertility data would be desirable, but that is no longer part of this work.

Materials and methods should specify that RUMI is a device that is placed on a collar.

Author Response

The work is very contemporary, focusing on precision dairy cattle breeding and the contributions RUMI can offer farmers in heat detection and optimal insemination timing. The most significant objection I find to the work is the second proposed objective: "To determine the optimal timing of artificial insemination using the device."

The methodology followed does not allow for determining the optimal timing of insemination, since artificial insemination is not performed. However, it does allow for establishing a recommendation for the optimal insemination interval, based on one's own observations and the results of other authors. I believe the objective should not be "TO DETERMINE" but rather "TO ESTABLISH a recommendation for the optimal insemination interval." The use of the device facilitates identifying this timing. Fertility data would be desirable, but that is no longer part of this work.

First, the authors would like to thank the reviewer for taking their time to revise the manuscript and make the necessary comments and suggestions.

Following the reviewer’s suggestions, the objective has been reformulated and the manuscript was modified in the abstract (L27) and objectives (L102) sections.

Materials and methods should specify that RUMI is a device that is placed on a collar.

Reviewer 2 Report

Comments and Suggestions for Authors

The study evaluated the performance of the neck-mounted accelerometer sensor system RUMI in estrus detection and determination of optimal insemination timing in Holstein cattle. My concerns are as follows:

  1. Supplement specific information about the sensor, including core parameters and a schematic diagram of the system's operation.
  2. The assessment of blood flow in dominant follicles and corpus luteum used subjective grading (0-3 levels), without stating inter-rater reliability tests (such as Kappa coefficient).
  3. Ultrasound examinations are performed every 12 hours, while changes in the ovaries and uterus during the estrous cycle may be more rapid. This frequency may lead to an estimation error of ±6 hours in ovulation time, thereby affecting the accuracy of the "interval from estrus onset to ovulation" and the "optimal insemination window".
  4. The article mentions that RUMI generates an estrus index through the combination of multiple behavioral parameters and AI algorithms, but the core logic of the algorithm (such as parameter weights and the basis for threshold setting) is not disclosed, making it difficult for other studies to replicate and verify the results.
  5. The study proposes that "11.4–15.5 hours after estrus onset is the optimal insemination time", but does not combine conception rate data to verify the effectiveness of this window.

Author Response

The study evaluated the performance of the neck-mounted accelerometer sensor system RUMI in estrus detection and determination of optimal insemination timing in Holstein cattle. My concerns are as follows:
1.    Supplement specific information about the sensor, including core parameters and a schematic diagram of the system's operation.
First, the authors would like to thank the reviewer for taking their time to revise the manuscript and make the necessary comments and suggestions. 
Specific information about the sensor was included in section 2.2 (L121-144).
2.    The assessment of blood flow in dominant follicles and corpus luteum used subjective grading (0-3 levels), without stating inter-rater reliability tests (such as Kappa coefficient).
All examinations were recorded and analyzed by the same person, with training and experience on the matter. Therefore, inter-observer agreement is not possible to calculate. This matter was clarified in the manuscript (L167-168).
3.    Ultrasound examinations are performed every 12 hours, while changes in the ovaries and uterus during the estrus cycle may be more rapid. This frequency may lead to an estimation error of ±6 hours in ovulation time, thereby affecting the accuracy of the "interval from estrus onset to ovulation" and the "optimal insemination window".
Examinations were performed every 12 hours to maximize animal welfare and optimize operators’ logistics. Considering that this study was conducted on a commercial farm, we also tried to interfere as less as possible on regular farm management. We acknowledge that this 12 h interval might pose a limitation and should be taking into account when interpreting the results. The manuscript was modified in the discussion (L357-360).
4.    The article mentions that RUMI generates an estrus index through the combination of multiple behavioral parameters and AI algorithms, but the core logic of the algorithm (such as parameter weights and the basis for threshold setting) is not disclosed, making it difficult for other studies to replicate and verify the results.
Additional information about the RUMI device was included in section 2.2 (L119-142). However, as this is a commercial product, further details cannot be disclosed.
5.    The study proposes that "11.4–15.5 hours after estrus onset is the optimal insemination time", but does not combine conception rate data to verify the effectiveness of this window.
The secund objective was reformulated to “establish a recommendation for the optimal timing for artificial insemination”, as we are just estimating the optimal artificial insemination window based on data previously published. Further studies should focus on determining the effectiveness of this window based on subsequent conception rates, as well as studying other possible alternatives.

Reviewer 3 Report

Comments and Suggestions for Authors

The manuscript, “Estrus Detection and Optimal Insemination Timing in Holstein Cattle Using a Neck-Mounted Accelerometer Sensor System”, presents a well-structured and methodologically sound study evaluating the performance of the RUMI neck-mounted accelerometer for detecting estrus and determining optimal insemination timing in dairy cows. The inclusion of ultrasonographic validation, hormonal profiling, and blood flow assessment provides a comprehensive approach rarely seen in similar studies.

The topic is relevant to precision livestock farming and reproductive management, and the manuscript provides clear practical implications for farm management. However, there are aspects that could be improved to enhance clarity, highlight novelty, and strengthen the discussion of results in relation to the broader literature.

Major Comments

  • While the use of accelerometer collars for estrus detection is established, the combination with detailed ovarian blood flow characterization and progesterone measurements is noteworthy. The Introduction could better emphasize this integration as a distinctive contribution.
  • The study identifies a narrow optimal insemination window (11.4–15.5 h after heat onset). This is an important finding, but it should be discussed in more depth in relation to variations in herd management, semen type (e.g., conventional vs. sex-sorted), and breed-specific factors.
  • The practical application of these results in commercial settings with different detection-to-insemination logistics should be addressed.
  • The use of one-way ANOVA is appropriate for the main comparisons, but the manuscript should explicitly state whether assumptions of normality and homogeneity of variance were tested.
  • Given the repeated measurements over time, a mixed-model approach could potentially provide more robust analysis for some variables.
  • The Discussion section is well structured but could more explicitly connect the findings on follicular blood flow to practical ovulation prediction strategies in field conditions.
  • The discussion should clarify how RUMI’s accuracy and specificity compare with other multi-parameter devices beyond accelerometer-only systems, and the potential reasons for differences.
  • Some potential limitations, such as the effect of the farm’s specific management practices, environmental factors, or collar placement accuracy, should be acknowledged.

Minor Comments

  • Abbreviations such as CL, AI, and P4 should be defined at first mention in the abstract.
  • Ensure consistent formatting of statistical values ( p ≤ 0.05) and units (h, ng/mL).
  • Consider shortening the Materials and Methods section slightly to avoid redundancy between subsections.

Author Response

The manuscript, “Estrus Detection and Optimal Insemination Timing in Holstein Cattle Using a Neck-Mounted Accelerometer Sensor System”, presents a well-structured and methodologically sound study evaluating the performance of the RUMI neck-mounted accelerometer for detecting estrus and determining optimal insemination timing in dairy cows. The inclusion of ultrasonographic validation, hormonal profiling, and blood flow assessment provides a comprehensive approach rarely seen in similar studies.

The topic is relevant to precision livestock farming and reproductive management, and the manuscript provides clear practical implications for farm management. However, there are aspects that could be improved to enhance clarity, highlight novelty, and strengthen the discussion of results in relation to the broader literature.

We would like to thank the reviewer for taking their time to revise the manuscript and make the necessary comments and suggestions to improve its quality.

Major Comments

  • While the use of accelerometer collars for estrus detection is established, the combination with detailed ovarian blood flow characterization and progesterone measurements is noteworthy. The Introduction could better emphasize this integration as a distinctive contribution.

Changes were made accordingly in lines 71-77

  • The study identifies a narrow optimal insemination window (11.4–15.5 h after heat onset). This is an important finding, but it should be discussed in more depth in relation to variations in herd management, semen type (e.g., conventional vs. sex-sorted), and breed-specific factors.

Changes were made accordingly in lines 364-366.

  • The practical application of these results in commercial settings with different detection-to-insemination logistics should be addressed.

This study was conducted using the RUMI monitoring devices, and therefore all the results reported should be only considered when using this device. A clarification was included at the end of the discussion, explaining that each monitoring device should be carefully validated and the information obtained from one device should not be extrapolated to other devices (L373-376).

  • The use of one-way ANOVA is appropriate for the main comparisons, but the manuscript should explicitly state whether assumptions of normality and homogeneity of variance were tested. Given the repeated measurements over time, a mixed-model approach could potentially provide more robust analysis for some variables.

Changes were made accordingly in statistical analysis section (L200-201).

  • The Discussion section is well structured but could more explicitly connect the findings on follicular blood flow to practical ovulation prediction strategies in field conditions.

Although we think that combining mode B and Doppler ultrasound to monitor cows in heat and near ovulation is useful for the objectives established in the manuscript, its use in field conditions might be impractical due to the need for repeated examinations. This is clarified in the discussion (L340-342).

  • The discussion should clarify how RUMI’s accuracy and specificity compare with other multi-parameter devices beyond accelerometer-only systems, and the potential reasons for differences.

The manuscript was modified according to the reviewer’s suggestions and information about comparisons with other devices was included in the discussion (L281-287).

  • Some potential limitations, such as the effect of the farm’s specific management practices, environmental factors, or collar placement accuracy, should be acknowledged.

Change was made accordingly in line 131 and 344-346.

Minor Comments

  • Abbreviations such as CL, AI, and P4 should be defined at first mention in the abstract. Ensure consistent formatting of statistical values ( p ≤ 0.05) and units (h, ng/mL).

Changes were made accordingly

  • Consider shortening the Materials and Methods section slightly to avoid redundancy between subsections.

The materials and methods section was modified accordingly and some information was removed.

Round 2

Reviewer 2 Report

Comments and Suggestions for Authors

The authors have adequately addressed all of my concerns. I recommend the manuscript for acceptance.